# Matrix Metalloproteinases and Their Inhibitors in the Pathogenesis of Epithelial Differentiation, Vascular Disease, Endometriosis, and Ocular Fibrotic Pterygium

**DOI:** 10.3390/ijms26125553

**Published:** 2025-06-10

**Authors:** Jun-Young Park, Yeonwoo Choi, Hee-Do Kim, Han-Hsi Kuo, Yu-Chan Chang, Cheorl-Ho Kim

**Affiliations:** 1Molecular and Cellular Glycobiology Unit, Department of Biological Sciences, Sungkyunkwan University, Suwon 16419, Republic of Korea; wnsdud2057@naver.com (J.-Y.P.); sakiz93@naver.com (Y.C.); hdk0330@naver.com (H.-D.K.); 2Environmental Diseases Research Center, Korea Research Institute of Bioscience and Biotechnology, 125 Gwahak-ro, Daejeon 34141, Republic of Korea; 3Department of Biomedical Imaging and Radiological Sciences, National Yang Ming Chiao Tung University, Taipei 112, Taiwan; hankkuo.be13@nycu.edu.tw

**Keywords:** MMPs, MMP inhibitors, TIMPs, epithelial cell differentiation, vascular disease, endometriosis, pterygium

## Abstract

Matrix metalloproteinases (MMPs) are key enzymes involved in the remodeling of the extracellular matrix (ECM) through the degradation of its components in a controlled endoproteolytic manner. Beyond ECM degradation, MMPs also target plasma membrane proteins implicated in signaling cascades and the progression of disease. Structurally, the catalytic function of MMPs is dependent on metal ions such as Zn^2+^. ECM remodeling by MMPs supports processes including tissue growth, morphogenesis, elongation, and adaptation to environmental changes occurring under both physiological and pathological conditions. These activities are subject to tight regulation by cellular MMP enzymes. While the current body of research has primarily centered on the functions of MMPs and their roles in cancer biology, knowledge of their involvement in vascular disease, endometriosis, fibrotic eye disease, epithelial cell differentiation, and the actions of MMP inhibitors remains comparatively sparse. This review explores the roles of MMPs in vascular disease and endometriosis, particularly as they relate to the ectopic growth of endometrial tissue. In addition, we summarize evidence regarding their contributions to disease mechanisms, with a focus on pathological progression. Due to their significant therapeutic promise in a variety of human diseases, advancing our understanding of MMP biology is likely to facilitate progress in clinical application and the development of novel interventions. This review also evaluates advances in the development and therapeutic potential of MMP inhibitors.

## 1. Introduction

Protease enzymes mediate the hydrolytic cleavage of peptide bonds within proteins. In vertebrates, 29 matrix metalloproteinase (MMP) genes display overlapping enzymatic activities. Genomic data from both vertebrates and invertebrates have facilitated the exploration of the diversity of proteases. Among 600 hydrolase genes, protease-encoding genes account for less than 2% of the entire human genome. MMPs constitute a subgroup of multi-domain Zn^2+^ ion-dependent endopeptidases. The activity of MMPs in living organisms is tightly regulated through a precise equilibrium between MMPs and their inhibitors. In essence, organisms maintain a balance between MMPs and their respective inhibitors. Disruption of the balance between MMPs and their inhibitors leads to pathological conditions including vascular, endocrine, and fibrotic diseases, as well as accelerated dermal aging. The fibrotic microenvironment is critically involved in the initiation and progression of fibrosis, primarily influenced by proteases through the regulation of inflammation, angiogenesis, and the innate immune response. Proteases further modulate intracellular signaling mechanisms, particularly those mediated by mitogen-activated protein kinases (MAPKs), affecting inflammation, immune function, and apoptotic cell death at both the cellular and subcellular levels. Consequently, certain proteases implicated in disease processes offer significant therapeutic potential for disease diagnosis and treatment.

Generally, among proteases, MMPs are involved in maintaining normal growth, senescence, dermal remodeling, and processes such as wrinkle formation during aging. MMPs facilitate direct tissue reorganization during the physiological implantation of embryonic fusion cells into the uterine environment and participate in tissue remodeling by enabling skeletal turnover and organ reformation. Furthermore, MMPs are implicated in various physiological and pathological phenomena. Upregulation of MMP expression has been reported in multiple human disorders, including arthritis, inflammation, periodontal lesions, pulmonary lesions, cardiac lesions, and tumor progression. In addition, MMPs are central to the pathogenesis and disease progression of arthritis, vascular disorders, inflammation, and cancer [1]. Given the multifaceted involvement of MMPs in human disease, this review aims to focus on selected MMP-related human diseases that have not been previously addressed. MMPs are associated with tumor formation, fibrosis, vascular disease, and rheumatoid arthritis. MMPs exhibit regulatory activities in overall pulmonary fibrosis, exerting both antifibrotic and pro-fibrotic effects. MMPs also contribute to scar resolution and facilitate effective regeneration following organ transplantation [2]. Moreover, MMPs possess therapeutic utility in the treatment of burns and ulcers [3]. Additionally, MMPs are directly implicated in vascular disease, endometriosis, ocular fibrotic disorders, and epithelial cellular differentiation.

Various strategies have been employed for the development of both naturally occurring and chemically synthesized protease inhibitors that are pertinent to human disease therapeutics. Selective MMP-targeting inhibitors have been developed; however, the high degree of homology in MMP catalytic domains complicates the distinction of these inhibitors. The most successful applications of MMP inhibitors are in inflammatory and vascular diseases, as multiple MMPs are implicated in collagenous colitis, diverticulitis, and disorders of the digestive system. Additionally, therapeutic MMP inhibitors are required to cross the central nervous system barrier, as their preventive potential is significant. Strategies for inhibitor development also incorporate stem cell and cancer stem cell approaches, in addition to targeting epigenetic regulators and non-coding microRNAs. In vivo direct-acting probes have also been adopted as bioimaging tools for disease diagnosis and monitoring.

Among proteases, MMPs, or metalloproteinases, are enzymes that degrade proteins and require active metal ions for their catalytic activity. There are over 20 distinct MMPs, which facilitate cell migration by breaking down components of the basement membrane and cell matrix. Expression of MMPs is commonly elevated in tumors and inflamed tissues. MMPs possess three highly conserved domains (pro, catalytic, and hemopexin domains). By means of these domains, MMPs contribute to tissue cleavage, the removal of physical barriers, and the release of signaling molecules, including proteases that disrupt the extracellular matrix (ECM). Thus, MMPs are responsible for the breakdown of most ECM components and play a critical role in both physiological and pathological processes throughout the body. Disease onset, as well as cell, tissue, and vascular remodeling, is regulated by exogenous MMP inhibitors or endogenous TIMPs, as the interplay between MMPs and their inhibitors is crucial for cell–cell recognition, immune function, differentiation, and development. Therefore, in this review, we focus on vascular thickening in development and progression, the emergence and advancement of endometriotic lesions in the uterine endometrium, eye fibrotic eye pterygium categorized as fibrovascular fibrosis, and epithelial cellular differentiation, as outlined in Figure 1. The state of the art in these domains, along with clinical trials investigating MMPs and their endogenous as well as synthetic or natural inhibitors directed against diseases at various stages, is discussed. Moreover, the effects of MMPs and MMP inhibitors in relation to inhibition-associated side effects are analyzed for their clinical implications. Collectively, this review highlights the diverse functions of specific MMPs in human disease, the roles of their inhibitors, and directions for future research.

## 2. MMPs as Biomarkers in Human Diseases

Collagen degradation mediated by epithelial-cell-derived MMPs is essential for epithelial cell differentiation. Beneath the epithelium, the ECM primarily consists of collagens, and this matrix is thicker in goblet cell hyperplasia compared to normal epithelial tissue. MMPs facilitate the degradation of ECM components, including collagens, across various tissues to modify the tissue microenvironment. In the respiratory mucosa under physiological conditions, MMPs function as enzymes critical for tissue remodeling, wound healing, and morphogenesis. MMPs produced by fibroblasts or inflammatory cells impact airway inflammation by regulating epithelial development and aberrant remodeling.

MMPs enzymatically cleave structural ECM proteins and are closely linked to inflammatory diseases, as these pathologies stem directly from MMP-mediated ECM protein degradation and remodeling. The human genome contains 24 MMP genes located on chromosomal DNA. Of these, two genes encode the identical MMP-23 enzyme, which results in 23 distinct MMPs being classified. While MMP enzymes are generally found in vascular arterial tissues, certain subtypes, such as MMP-2, -14, and -19, show specific expression in these tissues. MMP-2, -14, and -15 are also highly expressed in endothelial tissues. In contrast, most MMPs display weak expression in normal tissue [4]. Macrophages and endothelial cells produce and secrete TNF-α through intracellular signaling pathways involving MAPKs and extracellular-signal-regulated kinase (ERK)-1/2 at inflammatory sites [5,6]. Among inflammatory cytokines, tumor necrosis factor (TNF)-α stimulates MMP-9 biosynthesis in human fibroblasts, fibrosarcoma, and smooth muscle cells. MMP-9 is also markedly upregulated in gingival fibroblasts derived from aged rats compared to those from younger rats [7]. In a therapeutic context, MMP-9 inhibitors have been shown to promote re-differentiation during epithelial-mesenchymal transition (EMT). EMT is a reversible differentiation process commonly observed in cancer progression, embryonic stem cell development, and wound healing. Unlike normal epithelial cells that form polarized sheets, mesenchymal cells lack junctions with neighboring cells. During cellular differentiation, epithelial cells may undergo transformation into a mesenchymal phenotype characterized by reduced cell–cell adhesion and decreased epithelial biomarker E-cadherin, alongside increased mesenchymal biomarker *N*-cadherin, which together promote enhanced cell motility and facilitate ECM degradation via MMPs. MMP-9 facilitates EMT, thereby increasing the adhesion, invasion, and migration of vascular smooth muscle cells (SMCs). The inhibition of MMP-9 suppresses EMT as well as the proliferation, migration, and adhesive behaviors of vascular SMCs.

## 3. MMPs as Biomarkers in Epithelial Cell Differentiation

Airway epithelial cells secrete MMP proteins, which contribute to collagen degradation during epithelial development. Throughout epithelial cell differentiation, MMP enzymes derived from epithelial cells play crucial roles in remodeling the ECM of airway mucosal epithelial cells in both nasal mucosa and polyp mucosa. For instance, the proliferative and migratory capabilities of human lens epithelial cells are substantially enhanced by EMT processes. Transforming growth factor-β1 (TGF-β1) is also known to induce EMT in human lens epithelial cells [8]. Additionally, MMPs are involved in cataract formation by mediating the EMT process. MMPs contribute directly to the development of subcapsular cataracts through associated cellular morphological changes. Inhibiting specific MMP-9 activity disrupts typical epithelial differentiation. The gene expression of the MMP-9 subtype is initially stimulated by TGF-β treatment, and subsequently, MMP-2 expression is upregulated. Among the various MMPs, the MMP-9 subtype gene is categorized as an early-response gene, although both MMP-2 and -9 are induced by TGF-β in lens tissues. Of these, MMP-9 activity is essential for proper ciliated cell differentiation. Mucociliary development specifically depends on the activities of both MMP-9 and MMP-13. Broad-spectrum MMP inhibitors such as GM6001 as well as selective MMP-2/-9 inhibitors prevent the formation of cataractous plaques [9].

Membrane-anchored a disintegrin and metalloproteinase (ADAM) proteases on the cell surface are involved in cell migration, inflammation, and proliferation through signaling activation. ADAM17, ADAM9, and ADAM17 interact with surrounded immune cells. For example, when ADAM17 is associated with the development of atherosclerosis, it promotes inflammatory responses and regulates cellular oxidative stress, and consequently exacerbates pathological changes [10]. However, ADAM17’s intracellular function and role in an in vivo environment are not well explored in atherosclerosis, and the atherosclerotic action of ADAM17 is not clearly understood yet. Copper homeostasis has been suggested to be associated with the mechanism of atherosclerosis. For example, ADAM1 upregulates atherosclerosis through cellular copper homeostasis [11]. Excessive free copper ions can cause oxidative stress and damage cell membranes, thereby impairing the function of vascular endothelial cells, increasing the absorption of cholesterol in the arterial wall and the release of inflammatory factors, and thereby accelerating the development of atherosclerosis [11]. ADAM12 is also involved in the thrombin-exerted proliferation of vascular SMCs [12].

A broader range of MMPs have been implicated in epithelial differentiation. Notably, the concentrations of MMP-7, -9, -10, and -13 are modestly increased in a normal epithelium. ECM-degrading MMPs, including MMP-7, -10, and -13, display elevated expressions, especially under normal conditions. MMP inhibition disrupts typical ciliation and causes an increased proportion of goblet cells. A reduction in overall MMP levels is noted in goblet epithelial cell metaplasia within inflamed areas of the nasal mucosa. Distinct MMP family members with variable expression regulate the processes of collagen assembly and disassembly. The airway epithelium supports homeostatic mucosal respiration. Aberrant regulation of epithelial cells results in pathological tissue remodeling and promotes inflammatory responses in the airway mucosa. Increased collagen deposition is observed concurrent with elevated expression of MMP-7, -10, and -13 in ciliated cells. The levels of MMP-7 and -13 are higher in normal ciliated epithelia compared to epithelia undergoing goblet cell metaplasia. Furthermore, the expression of MMP-7, -10, and -13 is closely correlated with ciliogenesis in airway epithelial cells. Consequently, the inhibition of MMPs results in abnormal phases of cellular differentiation and an increase in goblet cell numbers.

## 4. Functional Roles and Expression of MMPs in Vascular Diseases

In 1973, Ross and Glomset [13] proposed the “response to injury” hypothesis to explain abnormal proliferation of vascular smooth muscle cells (SMCs) and impaired endothelial tissue responses following injury. In the arterial wall, three types of cells are present: (1) endothelial cells (ECs), (2) SMCs, and (3) adventitial cells. MMP expression in vascular disease has been examined in cases of human intimal thickening. In the progression of vascular disease, SMCs degrade the ECM through the secretion of MMPs. A minimum of 14 MMPs from among the mammalian MMPs, either as secreted or surface-bound proteases, are found in vascular cells. MMPs including MMP-1, -2, -3, -7, -8, -9, -12, and -13 are commonly identified in intimal thickening regions of human atherosclerotic lesions [5,6,14]. The enzymatic activities of MMP-1, -2, -3, and -9 have been detected via in situ zymogen detection techniques utilizing intimal-thickening tissues. Notably, the active form of MMP-8 is uniquely upregulated in regions of intimal thickening. Additionally, heightened activities of MMP-8 and MMP-9 have been demonstrated in the intimal thickening region, while increased activity of MMP-2 is specifically observed in the fibrous region. Moreover, the enzymatic activities of MMP-1, -3, and -9 were shown to correspond with their protein expression in SMCs [15,16,17], whereas both MMP-1 and -10 were found to co-localize within inflamed endothelial tissues. The expression of MMP-2 and -28 genes has been identified as a biomarker for intimal thickening of the endothelium, indicating an activated cell phenotype in blood vessels. Consequently, elevated activities of MMP-8 and -9 enzymes are considered significant risk factors for abnormally inflamed tissue in the vascular endothelium. Among the MMPs, the gelatinases MMP-2 and -9 are the primary types present in lesions of the arterial wall and are highly expressed in vascular rupture lesions and metastatic cancers. SMCs associated with MMPs offer potential targets for therapeutic strategies in arterial tissues. MMP-9 activity and expression are closely linked to the unstable progression of the intimal rupture phenotype. MMP-8 expression also contributes to the occurrence of endothelial plaque rupture [15]. In plaque lesions, MMP-2, -3, and -9 are significantly upregulated compared to their expression levels in normal tissues.

Among MMPs, MMP-9 is predominantly expressed in intimal thickening endothelial cells and SMCs in vascular diseases. MMP-1/9 expression requires the activation of the MAPKs ERK1/2 and p38, as well as phosphoinositide 3-kinase (PI-3k) [6]. MMP-9 expression is increased in human SMCs. Nerve growth factor and TNF-α modulate MMP-9 gene expression in SMCs through the ERK1/2 pathway [15], involving AP-1, NF-kB, Ets, SP-1, and retinoblastoma elements that interact with the MMP-9 5′-flanking regional promoter. The vascular cell cycle and its arrest are regulated by PTEN (phosphatase and tensin homolog), which represses AP-1/NF-kB-mediated MMP-9 gene expression in SMCs [6]. Platelet-derived growth factor (PDGF) and thrombin, in addition to TNF-α, also influence the enzymatic activity of MMP-9 in human SMCs. The region within 710 bp upstream in the 5′-flanking region of the MMP-9 gene promoter is regulated by TNF-α stimulation in human aortic SMCs, as the upstream ERK1/2 MAP kinase signaling pathway is involved in TNF-α-induced MMP-9 upregulation. The association between MMP-9-gene-related signaling, PDGF or TNF-α stimulation, and ERK1/2 MAPK is integral to SMC response [16]. Mitochondrial DNA damage in SMCs induces reactive oxygen species production and disrupts ATP synthesis. MMP-1 and MMP-3 expression also alters the SMC phenotype. Therefore, pharmacological approaches have been developed to inhibit MMP function in vascular SMCs, as summarized in brief (Figure 1).

## 5. MMPs in Pterygium as an Orphan Disease

Pterygium, which is considered an orphan disease, is marked by the proliferation of fibrous and connective tissue and involves pathophysiological expansion of limbal cells, inflammatory cell infiltration, epithelial fibrosis, angiogenic collagen synthesis, and remodeling of the ECM. Pathological fibrosis in surfer’s eye (pterygium) is defined by abnormal cellular proliferation, an anti-apoptotic phenotype, and excessive ECM deposition by fibroblasts in ocular connective tissue, which allows for its classification as a benign tumor. Pterygium features fibrosis within the eye where the ECM accumulates excessively due to ongoing chronic inflammatory processes. Ocular pterygium represents a fibrotic condition, characterized by ECM accumulation linked with increased synthesis of collagen type III [18]. Disruption of the ECM structure leads to the release of vascular endothelial growth factor (VEGF), cytokines, and fibroblast growth factor (FGF), which drives migratory angiogenesis and the proliferation of pterygium cells. MMP-3 contributes to angiogenesis through its role in generating angiostatin. Thus, pterygium is categorized as a fibrotic disease affecting the ocular surface. The development of pterygium is intimately linked to the extracellular microenvironment homeostasis of ocular epithelial tissues, influencing its progression. Collagen type III is the predominant ECM component in pterygium, with smaller amounts of collagens type I and II [19].

### 5.1. MMPs in Intracellular Signaling of Pterygium

TGF-β1 serves as a crucial mediator in the fibrotic process within human corneas, facilitating myofibroblastic differentiation. The latent form of TGF-β1 binds to integrin αvβ8, and the active form of TGF-β1 is released through the activity of MMP-14 [19]. In pterygium, ECM remodeling is driven by MMPs secreted by myofibroblasts, while TGF-β1 further augments fibrotic contraction. MMPs play a significant role in both the development and prognosis of pterygium by directly promoting its advancement. Elevated levels of MMPs are observed in normal corneal, subconjunctival, and conjunctival fibroblasts. The head and body regions of pterygium also exhibit notable MMP expression. Consequently, MMPs are fundamentally involved in the progression of pterygium and its associated lesions. Furthermore, genomic markers related to MMPs are considered valuable as predictive indicators for the development and recurrence of pterygium. Subconjunctival, head, and body fibroblasts from pterygium display high expression of two distinctive forms, MMP-1 and MMP-3 [20]. During pterygium progression, fibroblasts localized to the lesion show significantly upregulated MMP-9 expression [21]. MMP-8 also contributes to the advancement of pterygium. Additionally, the co-expression of MMP-2 and MMP-9 is markedly enhanced in pterygium fibroblasts compared to normal fibroblasts. As such, MMP-2 and MMP-9 are tightly linked to the progression of pterygium in ocular diseases [22]. Expressions of MMP-2 and MMP-9 are absent in early-stage pterygium and in cultured fibroblasts. By contrast, increased levels of MMP-2 and MMP-9 are observed in fibroblasts from pathologically advanced stages of pterygium. However, the expression of MMP-3 and MMP-13 is downregulated in pterygium fibroblasts, accompanied by reduced cellular proliferation and migratory capacity [23]. The principal upstream regulator for intracellular signaling involved in fibrotic accumulation and synthesis is TGF-β1. TGF-β1 initiates signaling pathways that promote fibroblast proliferation and migration through EMT-mediated transdifferentiation into myofibroblastic cells [8]. Integrin αvβ8 bound by TGF-β1 stimulates MMP-14 expression (Figure 2).

Among the MMP family, the elevated expression of MMP-14 in pterygium leads to the degradation of abnormally accumulated collagen, despite MMP-14’s capacity to degrade collagens type I, II, and III [24]. MMP-14 is responsible for degrading all collagen types I, II, and III that are produced in pterygium tissue, and this process supports cell migration during fibrosis [25]. Pterygia demonstrate increased levels of MMP-1, -2, -3, -7 (matrilysin), -8, and -9 [26]. Additionally, pterygium cells stimulate local fibroblasts and facilitate the cleavage of fibrillar collagen through the activities of MMP-1 and -3. In advanced pterygium, MMP-1 expressed by fibroblasts is localized at the borders of the Bowman’s layer and corneal epithelium. Other MMPs, including MMP-2, -3, -7, -8, -9, and -14, are also present within pterygium tissue [26]. Among the less abundant MMP types, MMP-1, -7, and -3 are found in human pterygium fibroblasts in vitro; however, MMP-1, -2, and -9 are specifically expressed in pterygium tissue [21]. Furthermore, MMP-2 and -9 are frequently detected at high levels in pterygium samples from human patients. MMP-3 can activate additional MMPs, such as MMP-1, -9, and -13. Therefore, MMP-13 is involved in modulating both the inflammatory response and angiogenesis. Moreover, MMP-13 induces VEGF-A secretion from endothelial and fibroblast cells, facilitating collagenolysis-dependent angiogenesis and corneal vascularization [27]. Among the active MMP isoforms, ADAMs have also been implicated in human diseases. In the context of pterygium pathology, the upregulation of MMP-14 and ADAM-9, -10, and -17 is observed in human pterygia [28].

### 5.2. Prospective for MMP Inhibition in Pterygium Therapy

From a therapeutic perspective regarding pterygium development and progression, non-toxic antifibrosis compounds inhibit pterygium growth and help reduce recurrence after surgical excision. The primary treatment for pterygium remains surgical excision; however, recurrence frequently occurs following the procedure. Mitomycin C, a known chemotherapeutic agent, exhibits antifibrotic effects in cases of clinically recurrent pterygium. Nonetheless, complications of mitomycin C include ocular tissue damage that may lead to corneal melting [29]. As a result, the development of effective therapies to prevent and manage recurrent pterygium remains a significant goal. The endogenous TIMP-1 inhibits MMP-1 activity through co-expression, acting as an MMP-1 inhibitor, but increased MMP-1 disrupts the balance with TIMP-1. Disruption in the equilibrium between MMP-1 and TIMP-1 contributes to pterygium invasion into corneal cells and tissue [30]. For instance, an MMP-3 inhibitor suppresses angiogenesis due to its broad-substrate specificity. Selective inhibitors of MMP-2, -9, and -14 are being explored as potential options for pterygium therapy. Notably, (R)-ND-336 is recognized as a selective inhibitor of MMP-2, -9, and -14 and demonstrates well-characterized mechanisms of action. However, (R)-ND-336 acts as a noncompetitive inhibitor for MMP-8. Currently, there is no documentation of inhibitors targeting MMP-1, -3, or -7, or ADAM9 and ADAM10. Therefore, (R)-ND-336 is regarded as a candidate therapeutic inhibitor of MMP-14 in human pterygium. (R)-ND-336 reduces the migratory capacity and collagen accumulation by human conjunctival fibroblasts in pterygium. This small-molecule inhibitor blocks MMP-14 activity as well as collagen synthesis and migration in conjunctival fibroblasts characteristic of pterygium formation. (R)-ND-336 shows promise as a therapeutic agent for pterygium [28]. Likewise, pirfenidone (5-methyl-1-phenyl-2-[^1^H]-pyridone) demonstrates antifibrotic activity in clinical trials and suppresses TGF-β, PDGF, connective tissue growth factor, and TNF-α signaling pathways [31]. As both an anti-inflammatory and antifibrotic compound, pirfenidone efficiently suppresses hepatic stellate, intestinal fibroblastic, leiomyoma, myometrial, and renal cells. Pirfenidone inhibits TIMP-1 and TGF-β signaling after glaucoma surgery. It also reduces fibroblast cell migration and proliferation in the conjunctiva through the TGF-β/MMP-1/TIMP-1 axis. Among monoclonal antibodies, bevacizumab, an angiogenesis inhibitor, prevents VEGF from binding to VEGFR on endothelial cell surfaces. Subconjunctival injection of bevacizumab reduces recurrence rates of pterygium in patients [32]. Bevacizumab also suppresses MMP-1 expression in Tenon’s fibroblasts derived from recurrent pterygium [33] and inhibits MMP-3 and -13 in in vivo pterygium tissue.

## 6. MMPs in Endometriosis

Endometriosis frequently affects menstruating females and is characterized by an inflammatory state prior to disease progression. Endometriosis refers to the abnormal proliferation of endometrial tissue outside the uterus. MMPs constitute a significant aspect of the pathophysiology and etiology of endometriosis, as their production predominates in endometriotic tissue and influences both the development and progression of the disease. Autoantibodies targeting endometrial and serum antigens are specifically observed in endometriosis, which is associated with its nature as a chronic, complex inflammatory disorder. Within tissue microenvironments, MMPs are implicated in disrupting immune homeostasis and contributing to related diseases via immune cell modulation or the regulation of autoimmune factors. The expression levels of MMPs as biomarkers are altered in the endometrial tissues of patients with endometriosis. Fourteen MMPs are present in human vascular endothelial cells. Tissues from patients with endometriosis demonstrate abnormal expression of MMP-1, -2, -3, -7, -9, -10, -11, -12, -13, -23, and -26, as well as MT1-MMP and MT5-MMP [34]. Specifically, the eutopic endometrium exhibits increased expression of MMP-1, -23, MT1-MMP, and MT5-MMP, whereas the ectopic endometrium shows the upregulation of MMP-2, -3, -7, -9, -10, -11, -12, -13, -23, -26, and MT1-MMP [35]. Notably, MMP-7 is absent in endometriotic tissue but present exclusively in epithelial endometrial cells. In contrast, MMP-23 and MMP-26 are highly expressed in endometriotic tissue, although their precise functional roles remain unclear.

MMPs present potential applications in the diagnosis and targeted treatment of endometriosis. The interplay between MMPs and TIMPs is fundamental to the normal physiological function of the endometrium. In women, endometrial tissue exhibits an imbalance in TIMP and MMP expression and stoichiometry, facilitating infiltration and tissue remodeling. Maintaining a normal MMP:TIMP ratio helps prevent or normalize invasive endometriosis development. MMPs in the endometrium are regulated by growth factors (GFs), cytokines, and steroid hormones. Aberrant expression of MMPs can initiate endometriosis, as the disease is defined by ectopic growth of endometrial tissues. Consequently, MMPs represent promising therapeutic targets for endometriosis. The Thomsen–Friedenreich antigen (Galβ1–3GalNAc), categorized as a type 1 core *O*-glycan, is detected in the serum of endometriosis patients and binds to specific ligands [36]. T antigens found on MMP-9 and glycosyl pro-MMP-9 multimers interact with sera from individuals with endometriosis [37]. The involvement of MMPs in endometriosis etiology is partly mediated by anomalous responses to cytokines and GFs within the eutopic endometrium associated with the disease (Figure 2).

MMPs play a pivotal role in endometriosis’s invasive behavior, with cytokines contributing to the development of endometriosis. Cytokines upregulate MMP expression within endometrial tissue. Specifically, TNF-α and IL-1β promote the gene and protein expression of MMPs in endometriosis tissue, thereby facilitating the development and progression of the disease. The earliest stage of progression involves the growth of ectopic endometrial tissue characteristic of endometriosis. Anti-TNF-α therapy acts to inhibit MMP transcription due to its downstream pathways, as both IL-1β and TNF-α serve as inducers of MMP expression in endometrial and endometriotic tissues. Endometriotic lesions display an elevated expression of MMP-1, -2, -3, -7, and -9, which is associated with disease development and progression and enhances endometriotic tissue adhesion and invasion. The adhesion of endometrial tissue stimulates MMP expression in endometriotic lesions. Subsequently, MMPs degrade the ECM to support stable vessel formation and angiogenesis, thereby sustaining endometriosis. The use of MMP inhibitors could be advantageous for the treatment of endometriosis. Nevertheless, data on MMP inhibitors in the management of endometriosis remain scarce. Overall, the information regarding anti-MMP therapies in endometriosis is both limited and insufficient [38]. Anti-MMP therapies remain under consideration in the context of endometriosis treatment.

### 6.1. MMP Signaling in Endometriosis

MMPs contribute to cancer cells’ ability to evade host immune surveillance by enzymatically cleaving cell surface ligands of the NK group 2 member D (NKG2D) [39]. IL-10 and IFN-γ upregulate the expression of MMP-2 and -9, while IL-6 specifically increases MMP-9 expression, and IL-1β modulates MMP-13 activity [40]. TNF-α induces the expression of MMPs in endometrial tissues while simultaneously reducing the levels of host tissue TIMPs [41]. IL-2 and IL-27 collectively downregulate MMP-9 expression through the regulation of IFN-γ and IL-10, thereby influencing the invasiveness of endometriosis cells [42]. IL-34 enhances MMP-9 expression in endometriosis, whereas IL-37 modulates MMP9 expression in endometrial cells [43]. MMP-2 and -9 act as downstream biomarkers involved in the progression of endometriosis. Moreover, both estrogen and cytokines stimulate the expression and activity of MMPs. Estrogen specifically increases MMP-9 expression, and aquaporin 1 upregulates both MMP-2 and -9. MMP-2 expression is also regulated by the Cox-2/PGE2/pAKT axis as well as the leptin/JAK2/STAT3 axis. Beyond its protease functions, the MMP-7-EGFR signaling pathway modulates the EMT process, which is essential for endometriosis development and progression [44]. Additionally, MMP-7 expression is increased as a downstream effect of EGFR signaling. In endometriosis, metal iron facilitates EMT and enhances the activities of MMP-2 and -9. MMPs in the endometrium are essential for regulating endometrial thickness and structure. In both eutopic and ectopic endometrium, endometrial tissues can respond to hormones in processes such as EMT, migration, invasion, angiogenesis, and fibrosis, although in endometriosis, these responses become dysregulated. In relation to fibrosis, MMPs promote collagen accumulation in endometriosis-associated fibrosis. Increased fibrosis results from heightened synthesis of type I and type III collagen. Additionally, the suppression of MMP-2 and -14 expression leads to reduced fibrosis levels. Silencing MT1-MMP increases fibrosis through enhanced collagen accumulation, while greater matrix stiffness further boosts collagen I synthesis and the expression of MMP-1/MMP-14. Thus, the collagenases of the MMP-1, -8, and -13 families participate in angiogenesis. MMP-1 stimulates VEGF receptor-2 (VEGFR2) synthesis, and MMP-7 promotes angiogenesis via the VEGF-VEGFR2 pathway [38].

### 6.2. MMP and TIMP Regulation in Endometriosis

Remarkably, although MMPs regulate the pathophysiology and development of endometriosis, TIMP-1 and -2 expression levels are decreased in this condition. In endometrial cells, α2-6 sialylation in proteins promotes the progression of endometriosis [45]. When endometrial cells are treated with progesterone or TIMP-1, it has been established that MMP expression is suppressed. Given that MMPs play crucial roles in the initiation and progression of endometriosis, endometrial stromal cells facilitate migration through the remodeling of collagen type I, enabling the invasion of endometrial cells. MMP-14, which is recognized as the first MT1-MMP, is responsible for degrading the collagen I component of the ECM. In ovarian endometriosis, reductions in the cell polarity marker Par3 and the tight-junction protein occludin result in decreased epithelial cell polarity and weakened cell–cell tight junctions. Therefore, it should also be noted that MMP-7 contributes to EMT, and epidermal growth factor (EGF) stimulates the expression of MMP-7 (Figure 2) [44].

## 7. Exploration and Development of MMP Inhibitory Agents as Candidates for Preventive and Therapeutic Drugs Targeting Vascular, Endometriosis, and Pterygium Diseases

Based on their background, MMP inhibitors have emerged as promising therapeutic agents for human diseases such as vascular thickening and endometriosis and fibrotic eye disorders like pterygium. As previously described, elevated MMP levels and enzymatic activities are implicated in the pathogenesis of angiogenesis-related abnormal tissue remodeling, making MMPs a viable target for therapeutic intervention. The inhibition of MMPs is achieved through various mechanisms, including the suppression of MMP mRNA transcription, prevention of proMMP synthesis, blockade of proMMP secretion, interruption of proMMP activation to MMP, and direct inhibition of MMP enzymatic activity. Blocking MMPs may effectively limit the vascular infiltration of SMCs, given the critical pathophysiological roles MMPs fulfill in the human body. Therefore, there is a demand for anti-vascular MMP inhibitors exhibiting both selectivity and safety. MMP inhibitory agents were initially developed with broad-spectrum specificity, primarily by targeting the Zn^2+^ ion located within the catalytic site [46].

### 7.1. Computational and Artificial Intelligence (AI)-Driven Approaches for the Development of MMP Inhibitory Agents Targeting the MMP S1 Pocket-Binding Hydrophobic Group

Computational chemistry approaches have also been integrated with AI and machine learning methodologies. For instance, these MMP-1-inhibitory agents include methyl rosmarinate derivatives [47]. In a similar context, selective MMP-1 inhibitory agents have been synthesized using kinesin-like protein 11 (KIF11) mimicry analogs through structure–activity relationship (SAR) techniques [48]. Thiazole derivatives demonstrated inhibitory effects on MMP-1, -8, and -9 enzymes [49]. Regarding MMP-2 inhibitors, low-molecular-weight compounds specific for the MMP-2 type are considered promising candidates for preclinical evaluation. Through 4D-QSAR and pharmacophore modeling, β-N-biaryl-containing compounds such as sulfonamide hydroxamate analogs have been identified as possessing MMP-2 inhibitory properties [50]. Sulfonamide-based and dihydropyrazole-based sulfonamide analogs containing a 2-hydroxy phenyl group have shown MMP-2 inhibition [51]. Hydroxyquinoline and hydroxynaphtyridine derivatives represent selective inhibitors for MMP-2 and -13. These molecules structurally feature a zinc recognition group and are reported to lack adverse effects. Salicylaldehyde acts as a dual inhibitor for MMP02 and -8 [52]. 8-hydroxyquinoline derivatives demonstrated inhibition toward MMP-2 and -9, without manifesting pro-invasive or pro-angiogenic activities. Triazoles containing sulfonamide bridges have demonstrated activity against MMP-2 and additional MMPs [53]. Aryl triazolyl methyl aziridines exhibit selective inhibition of MMP-2 in melanoma and fibroblast cell lines [54]. For non-zinc-binding MMP-2 inhibitory agents, benzimidazole scaffolds serve as selective MMP-2 inhibitors [55]. To develop inhibitors capable of crossing the blood–brain barrier (BBB), gelatinase MMP-2 and -9 inhibitory agents have also been produced by Iproteos Co. Ltd., Spain [56]. For MMP-3 inhibition, the hydrophobic pocket within the MMP-3 catalytic site has been targeted by inhibitors that include zinc-binding motifs with a hydroxamate group, classified as hydroxamate-based MMP-3 inhibitors [57]. Through in silico docking analyses, 2-phthalimidinoglutaric acid derivatives have been identified as inhibitors for several MMPs, including MMP-3. Sulfonamide-containing dehydroabietic acid analogs are also identified as promising MMP-3 inhibitors. In the case of MMP-7 inhibitors, the MMP-7 and MMP-1 structures share S1′ structural features, which has presented challenges in developing highly selective inhibitors. Nitro-based MMP inhibitors selectively inhibit MMP-7 over MMP-1 [58].

For the screening of MMP inhibitors, computational 3D-QSAR pharmacophore models have been effectively used to identify natural compounds with inhibitory activity. The pharmacophoric model employed for the validation of natural products has also undergone experimental assessment for their binding to MMPs. These experimental evaluations focus on features such as aromatic rings, hydrogen bond donors and acceptors, and hydrophobic groups. Through a fragment molecular orbital (FMO)-based quantum mechanical approach, the dissociation constant of natural MMP inhibitors can be quantitatively assessed using a binding-affinity-related pharmacophore model method [59]. The hydrophobic group plays a vital role in facilitating interactions with the S1 pocket of MMP enzymes [60]. Recent data regarding MMP-1, -2, -3, -7, -9, -12, and -14 inhibitors have been retrieved from various databases for development purposes.

### 7.2. Chemical Synthetic MMP Inhibitors

Chemically synthesized inhibitors have been developed using combinatorial chemistry as inhibitory agents for MMPs. Furthermore, structure-based mimetic inhibitors have also been created to serve as broad-spectrum inhibitors. Although preclinical and clinical studies have demonstrated their significant therapeutic potential for vascular diseases, these broad-spectrum MMP inhibitory agents have not received approval from regulatory authorities such as the US FDA. To address the challenges associated with broad-spectrum MMP inhibitors, selectively inhibiting MMP inhibitors have been developed. For instance, the RS-130830 MMP inhibitor was found to worsen rupture-prone plaque formation in the endothelium [61]. Similarly, the broad-substrate-spectrum GM6001/ilomastat MMP inhibitor has been reported to reduce intimal thickening in experimental animal models [62]. Likewise, the CP-471,474 MMP inhibitor exhibits selective targeting towards MMP-2, -3, -9, and -13, but does not affect MMP-1. Oral administration of PG-116800 was evaluated in phase II clinical trials involving vascular patients. Compared to non-selective MMP inhibitors, selectively acting MMP inhibitors do not target all MMPs but act on specific members. Nevertheless, MMP inhibitors have yet to be approved for use in human diseases, whether in broad-spectrum or selectively acting forms. The inhibition of MMP-9 has been shown to modify vascular dysfunction [63], while RXP470.1, a selective MMP-12 inhibitor, has been reported to worsen abnormal vascular dysfunction [64]. MMP-12 inhibitors are thought to be advantageous due to their targeted disease-associated activity. The selective MMP-2 inhibitor, TISAM, has been shown to improve survival following the treatment of mice in an experimental vascular disease model. For MMP-12 inhibitors, there is ongoing investigation into their potential across various therapeutic areas. To date, no MMP-12 inhibitors have demonstrated efficacy in preclinical or clinical trials. Owing to the high degree of structural similarity within the MMP family, selective MMP-12-inhibiting agents remain undeveloped. Regarding MMP-14 inhibitors, *N*-isopropoxy-arylsulfonamide hydroxamate compounds have demonstrated binding to both MMP-9 and-14 in docking simulation studies [65]. NSC405020 has been developed as a hemopexin-domain-specific small molecule [66]. Isatin derivatives are documented to inhibit both MMP-9 and -14.

For synthetic MMP-9 inhibitors, multiple inhibitory agents have been developed to specifically target the MMP-9 catalytic site, but progression in clinical trials has been hindered by toxicity and low specificity. Notably, these MMP-9 inhibitory agents not only interact with the MMP-9 zinc ion but can also bind other metal ions present in different proteins. Batimastat (BB-94), a synthetic and peptidomimetic compound with a hydroxamate moiety, was the first MMP inhibitor advanced into clinical trials [67]; however, its poor solubility, suboptimal selectivity, mild toxicity, and adverse events such as abdominal pain limited further application. Marimastat (BB-2516), a synthetic low-molecular-weight inhibitor containing a hydroxamic acid moiety, is more suitable for oral administration. Its efficacy in preclinical and phase III clinical trials, especially in animal models of breast and lung metastasis, led to its progression to clinical evaluation [68]. CGS-27023A (MMI-270), a small-molecule and orally available inhibitor, is classified as a sulphonamide-type MMP inhibitor with broad-substrate specificity. Unlike peptidomimetic inhibitors, CGS-27023A demonstrated the inhibition of tumor growth in preclinical studies [47], but ultimately did not succeed in phase I clinical trials. Another broad-substrate inhibitor, the hydroxamate derivative CGS-25966, failed pharmacokinetically due to its adverse effects, such as musculoskeletal symptoms. Tanomastat (BAY12-9566), an orally available inhibitor with biphenyl and carboxylate groups functioning as a catalytic Zn^2+^ chelator, was well tolerated by ovarian tumor patients in a phase III clinical trial. Prinomastat (AG-3340), a nonpeptidic collagen-mimicking inhibitor, was assessed both orally and intraperitoneally in phase III clinical trials [69]. However, further development as an antitumor agent was discontinued due to the lack of efficacy and toxicity observed in clinical trials. The histone deacetylase inhibitor (HDAC) valproic acid downregulates MMP-2 and MMP-9 expression, reduces stem-cell-associated properties and tumor metastatic potential, and promotes the anaplastic redifferentiation of tumor cells. Sevoflurane inhibits miR-155 expression and reduces the expression of MMP-2 and MMP-9, which in turn limits tumor cell migratory invasion and induces apoptotic cell death in tumor cells [70]. Celecoxib, a cyclooxygenase-2 (COX-2)-selective inhibitor, decreases both the protein and mRNA levels of COX-2 and MMP-9. Furthermore, non-thermal atmosphere pressure plasma (NTP) has been shown to suppress human adult stem cell proliferation and tumor growth. NTP also reduces MMP-2 and -9 activity, as well as uPA activity, through the Akt/ERK pathway, and it modulates the cytoskeletal architecture via the FAK/Src axis, thereby inducing cellular morphological changes [71].

2-[[(4-Phenoxyphenyl)sulfonyl]methyl]-thiirane (SB-3CT), a selective synthetic MMP-2/-9 inhibitor, suppresses inflammatory endothelial angiogenesis by inhibiting MMP-9/VEGF-C expression [72]. In terms of pharmacokinetics, although SB-3CT exhibits low water solubility, it is metabolized to p-hydroxy SB-3CT via terminal phenyl oxidation, resulting in effective MMP-9 inhibitory activity. The development of multi-targeting MMP-9 inhibitors designed to selectively target two or three MMPs is an emerging trend in the functional enhancement of drug candidates. For instance, the oral inhibitor AZD1236, which specifically targets MMP-9 and -12, has been synthesized and applied therapeutically for spinal cord injuries and edemas. AZD1236 is also being investigated as an adjuvant therapy in patients with chronic obstructive pulmonary disease (COPD) [73]. The inhibitor AZ11557272, which is selective for MMP-9 and -12, reduces inflammatory responses, serum TNF-α production, and COPD severity [74]. Another oral inhibitor, AQU-118, targeting MMP-2 and -9, has demonstrated efficacy in reducing allodynia and dorsal root ganglion (DRG) lesions, making it a promising candidate for the treatment of neuropathic pain [58]. Although MMPs are promising therapeutic targets in human diseases, further clinical trials are necessary to advance their development [75].

### 7.3. Natural MMP-9 Inhibitory Agents

For MMP-9 inhibitory agents, natural MMP inhibitors have been investigated using a range of discovery tools. Numerous natural compounds that inhibit MMP-9 face limitations related to a high molecular weight and melting point, limited water solubility, instability, and low gastrointestinal permeability. For instance, recent assessments of natural MMP-9 inhibitors have examined their binding interactions with MMP enzymes. Developing and designing selective MMP-9 inhibitors may contribute to anticancer therapy with reduced adverse effects. The amino acids in the S1′ pocket are crucial for MMP-9 selectivity, as the hydrophobic aryl group fits the S1′ pocket and acts as a selective MMP-9 inhibitor [76]. For instance, MMP-9 inhibitors are regarded as promising anticancer targets. MMP-9 inhibitors displaying diverse structural frameworks have been identified. Most of these compounds lack MMP-9 selectivity and sufficient therapeutic efficacy due to the structural homology shared among various MMP enzymes. As a result, these MMP inhibitors have not received FDA approval for tumor therapy. AI-driven computational approaches have led to the identification of potentially effective MMP-9 inhibitors. The ligand-specific pharmacophore method has also been applied to natural MMP-9 inhibitors [77]. For example, the Guner–Henry scoring approach has served as a pharmacophoric model that includes both hydrogen bond acceptor sites and aromatic ring regions. This screening of natural compounds enabled the identification of MMP-9 inhibitory agents. Through molecular and dynamic docking simulations, an MMP-9 inhibitor was identified based on its binding free energy. Additionally, a combined method utilizing molecular dynamic simulation with quantum and molecular mechanics (QM/MM) calculations determined the binding characteristics at the MMP active site.

Regarding plant-derived MMP-9 inhibitors, epigallocatechin-3-gallate (EGCG) has been shown to suppress MMP-9 levels upregulated by vascular ischemic injuries and neurodegenerative damages in hippocampal CA1 and CA2 regions [78]. Mangiferin, a plant-derived polyphenolic compound sourced from *Mangifera indica*, inhibits MMP-9 both at the transcriptional and protein levels by suppressing NF-κB/AP-1 activities and increasing the expression of microRNA-15b (miR-15b) [79]. Mangiferin reduces β-catenin signaling and reverses epithelial–mesenchymal transition (EMT), as depicted in Figure 1 and Figure 2. The phytosterol β-sitosterol and the stilbenoid resveratrol (RES) inhibit MMP-9 via the ERK1/2/JNK1/2 MAPK pathways [80]. Several phenolic compounds, such as caffeoylquinic acid (known as chlorogenic acid), 5′-caffeoylquinic acid, rosemarinic acid (ROS), caffeic acid (CA), CA phenethyl ester (CAPE), CA ester, and 3,4-dihydroxyphenyllactic acid, are classified as zinc-ion-chelating MMP inhibitors [81,82]. Cinnamic acid analogs demonstrate therapeutic efficacy in vascular diseases, as computational drug discovery approaches assess the binding affinity of these analogs to the MMP-9 active site. Chlorogenic acid, cynarin, and rosmarinic acid are capable of targeting the MMP-9 catalytic domain at picomolar concentrations [83]. Quercetin, a polyphenolic flavonoid, demonstrates multiple inhibitory effects in cancer, diabetes, and viral infections, along with anti-oxidant activity. Quercetin also inhibits MMP-9 enzyme activity but enhances E-cadherin-mediated signaling, which, in turn, suppresses cell migration and EMT [84]. The hydrogen bonds and Zn-O coordination bonds of flavonoids (quercetin, myricetin, galbanic acid) are key inhibitory factors in their interactions with MMP-9 [83]. These flavonoids can bind to the hydrophobic pocket region formed by amino acids Ala189, Leu187, Ala 19, Tyr245, Pro246, and Met247, closely localized near the Zn atom in the MMP-9 enzyme. Molecular dynamics simulations have demonstrated that hydrogen bonding is essential for the interaction between galbanic acid and MMP-9 [83]. For cinnamic acid derivatives, inhibition occurs at the picomolar scale and is mediated through the catalytic Zn2+ ion and the S1 pocket of MMP-9, as confirmed by deep learning, molecular docking, and dynamic simulation analyses [83]. Utilizing these approaches, ginsenoside Rg1 has been found to suppress the migratory and invasive behavior of breast cancer cells by downregulating MMP-9 [85]. The structurally similar Rg3 is also reported to lower MMP-2 and -9 levels, thereby inhibiting lymph node metastasis and angiogenesis [86].

### 7.4. MMP Inhibitors Based on Multiple Molecular Targets of TIMP Domains

TIMP-based MMP inhibitors have emerged as a promising strategy for therapeutic development. In vitro-directed evolution, in combination with yeast surface display (YSD), enables protein engineering by exploiting the relationship between sequence, structure, and function. This approach utilizes protein scaffolds, such as MMPs modulated by TIMPs, for the design of inhibitors. The MMP-inhibitory spectrum of typical TIMPs can be modulated by engineering the N-terminal domains of TIMPs. *N*-terminal regions within TIMPs regulate protein folding and stability, which is crucial for maintaining MMP inhibitory activity. A directed evolution platform using YSD and the N-terminal peptidyl regions of TIMPs, such as TIMP-2, has facilitated the generation of selective MMP-9 and MMP 14 inhibitors. Full-length TIMPs employed as scaffolds can produce MMP-targeted inhibitors. Additionally, TIMP variants have been derived through the directed evolution of human TIMP-1 displayed on yeast. For instance, certain TIMP-1 variants have shown the selective inhibition of MMP-3, even though MMP-3 and -10 share high sequence and structural similarity. Targeted structural modifications within the N-terminal and C-terminal regions of TIMP-1 can enable selective interaction with MMP-3, while avoiding interaction with MMP-10. Similarly, the N-terminal peptidyl region in TIMP-2 that has an MMP-inhibiting part has served as a scaffold for generating inhibitors with high specificity and affinity. By employing N-terminal domain mutants of TIMP-2, variants specific to MMP-9 and -14 have been selected using a yeast surface display method [87].

### 7.5. MMP Inhibitors Based on Natural Products

Plant-derived MMP-9 inhibitors encompass both natural products and chemically modified derivatives sourced from phenolics, flavonoids, and phytosterols, which have shown therapeutic efficacy against ultraviolet-induced damage and tumors, as well as periodontal, neurodegenerative, and ophthalmic diseases. Both natural and synthetic MMP inhibitors have been demonstrated to suppress disease progression [88]. Natural products can protect against aortic aneurysm due to their inhibitory activity on MMP-2 and -9. Secondary metabolites and their derivatives, including flavonoids, play a role in vascular protection. Flavonoids, which are polyphenolics or pigments, are primarily derived from benzo-γ-pyrone structures via the phenylpropanoid pathway. This biosynthetic pathway converts phenylalanine into 4-coumaroyl-CoA, which feeds into flavonoid synthesis. The inhibitory potential of flavonoids has been evaluated using hybrid QM/MM calculations, molecular docking studies, and molecular dynamics simulations. Baicalin inhibits the activity of both MMP-2 and MMP-9 and is effective in reducing abdominal aortic aneurysm. Combined treatment with baicalin and t-PA leads to a decrease in MMP-9 levels, thereby reducing the incidence of ischemic stroke. Curcumin and rosmarinic acid are effective inhibitors of MMP-13 activity. Curcumin has also been reported to suppress MMP-2 and MMP-9 expression by inhibiting Rac1/PAK1 signaling. Similarly, other compounds, including apigenin, chrysin, luteolin, primuletin, and quercetin, have demonstrated inhibitory effects. The calculated binding free energies of flavonoids at the enzyme’s catalytic site correlate strongly with experimentally determined EC50 values. Green tea polyphenols include catechin derivatives such as epigallocatechin (EGC), epicatechin-3-gallate (ECG), and EGCG. Utilizing in silico molecular docking simulations, EGCG has been shown to interact with MMP-9 by binding to both the MMP-9 propeptide domain and the ECG galloyl group, as supported by its binding affinity values. EGCG has been reported to suppress periodontic inflammation, atherosclerotic lesions, and oxidative injuries resulting from ultraviolet (UV) exposure. In addition, EGCG inhibits tumor cell proliferation and invasion by modulating the ERK MAPK/Akt/PI-3K/NF-kB/AP-1 axis and subsequently suppressing MMP-9 synthesis and activity. EGCG also reduces the expression levels of MCP-1 and MMP-9 and downregulates the TLR-4/MAPK/NF-κB signaling pathway, contributing to the stabilization of vulnerable vascular plaques. The combined administration of EGCG and t-PA has been shown to reduce MMP-2 and MMP-9 expression in both infarcted brain regions and the BBB. Genistein is beneficial in the management of ruptured vascular lesions through its anti-inflammatory effects, stabilization of arterial plaques, and inhibition of MMP-9. Ginkgetin has been found to improve lipid profiles and regulate MMP activity. Icariin lowers the expression of CD147, collagen type I/III, and MMP-9 while enhancing TIMP-1 levels. Icaritin promotes collagen accumulation in atherosclerotic lesions by inhibiting MMP-1/proMMP-1 activity in the aortic tissue.

Similarly, flavonoids derived from propolis attenuate cardiac fibrosis, reduce type I/III collagen, MMP-2 and -9, and TGF-β levels and diminish vascular sclerotic plaques. Quercetin decreases MMP-2 enzyme activity in aortic lesions. Quercetin also modulates myocardial rupture and ischemic stress responses through Akt activation and the upregulation of connexins. Resveratrol reduces cerebral ischemic lesion size by restoring the balance between MMP9 and TIMP-1. Moreover, resveratrol regulates multiple cellular components, including Akt, caspases, cyclooxygenase 2 (COX-2), cytokines, MMPs, NF-κB, nuclear factor (NF), erythroid 2 like 2, Wnt, vascular cell adhesion molecule (VCAM), and VEGF. Kaempferol suppresses proMMP-2 activity, decreases MMP-9 expression, and lowers myocardial infarct lesion level. Nobiletin improves vascular function and downregulates the expression of MMP-2, MMP-9, and ischemic stroke severity. Proanthocyanidin exhibits inhibitory effects on inflammation, hypertension, platelet aggregation, and thrombosis, as well as providing hypocholesterolemic action. Furthermore, proanthocyanidin reduces levels of cardiac collagen biosynthesis, TNF-ɑ, TGF-β, and the ratio of MMP-2/TIMP-2 in aldosterone-induced high-salt hypertension rats. Pinocembrin preserves BBB integrity and alleviates ischemic lesion severity by reducing MMP-2 and MMP-9 levels. Salvianolic acid modulates aortic aneurysm progression through the inhibition of MMP-2 and -9.

Another natural product, the biflavone (I-3, II-3)-biacacetin, is synthesized by oxidative dimerization involving cerium ammonium nitrate and poly-substituted diaryl-1,3-diketone. In silico docking analysis demonstrates that (I-3, II-3)-biacacetin interacts with MMP-9 through hydrophobic contacts at the S10 active site [89]. A specific MMP inhibitor, JNJ0966, has been discovered and isolated from bacterial compounds that interact with recombinant MMP-9 protein. JNJ0966 suppresses PDGF-induced SMC proliferation and ECM production by modulating MMP-9. JNJ0966 prevents the conversion of MMP-9 zymogen to its active enzyme and proenzyme to the MMP-9 catalytic form. JNJ0966 occupies a cavity at the cleavage site Arg-106 in the MMP-9 zymogen, which is distinct from the catalytic site [90].

Fatty acids have been known to downregulate the gene expression of inflammatory mediators including cytokines, chemokines, COX-2, NOS, and MMPs. Fatty acids such as eicosapentaenoic acid, decosahexaenoic acid, oleic acid, linoleic acid isomer, punicic acid, linoleic acid, α/γ-linolenic acid, elaidic acid, and cis- and trans-parinaric acid also inhibit the enzyme activities of MMP-2 and MMP-9 at micromolar-ranged Ki values [91,92]. The MMP inhibition of fatty acids depends on fatty acid chain length [92]. Hydroxyecdysones and essential fatty acids isolated from quinoa (*Chenopodium quinoa* Willd.) inhibit MMPs’ enzymatic activities [93]. As an action mechanism, MMP inhibitors bind to the enzymatic catalytic sites with weak binding selectivity due to catalytic site similarities with other MMP enzymes. 3-(E-3,4-dihydroxycinnamaoyloxyl)-2-hydroxypropyl 9Z, 12Z-octadeca-9, 12-dienoate isolated from *Cymodocea serrulata* has been known to bind to the unconserved MMP2 domain [94].

### 7.6. MMP-Targeting miRNA and Monoclonal Antibody (Mab)-Based MMP Inhibitors

MicroRNAs (miRNAs) belong to the class of non-coding RNAs, which are transcribed from genes but are not translated into proteins, functioning entirely at the RNA level. miRNAs are single-stranded RNAs, approximately 22 nucleotides in length. miRNAs target tumor suppressor genes or their promoters to regulate tumor progression. For instance, MiR-205, miR-214, and miR-203 inhibit MMP-9 expression, angiogenesis, and EMT. Conversely, miR-106a-5p promotes MMP-9 expression as well as the migration and invasion of tumor cells [95]. Long non-coding RNAs (lncRNAs), such as LncRNA UCA1, which are over 200 nucleotides in length, contribute to disease progression by upregulating MMP-9 expression [96]. Furthermore, levels of circular RNAs (circRNAs) such as circRNA DOCK1, produced through exon circularization, are elevated in tumor cells and facilitate the accumulation of MMP-9 in cancers [97].

MMP-9 and MMP-14-specific Mabs have been evaluated in cancers such as breast tumors and gastric and gastroesophageal junction adenocarcinoma and in autoimmune disease like ulcerative colitis [98]. Unlike chemical compounds, Mabs can inhibit MMP enzymes more specifically and generally exhibit fewer side effects and lower cytotoxicity than other MMP inhibitors. However, the limitations of Mabs include poor penetration into tumor cells and high production costs. While no selective ADAM9, ADAM10, or ADAM17 inhibitors have been developed to date, ADAM17-specific Mabs, such as D1(A12), have been created to target the catalytic site in in vitro and in vivo studies [99]. The Mab D1(A12).45 D1(A12) blocks the interaction between integrin and the disintegrin region of the ADAM17 protein in in vitro studies. Moreover, the MED13622 Mab also selectively binds to the ADAM17 protein, inhibiting the growth of migratory tumor cells and SMCs in preclinical studies. An MMP-9-specific monoclonal antibody inhibitor, andecaliximab, has shown efficacy in clinical trials.

## 8. The Representative Development Status Under Clinical Trials of Human MMP Therapeutics

Currently, several MMP-9 inhibitors have advanced to phase 1 and phase 2 clinical trials but have not demonstrated significant efficacy. In a randomized clinical trial involving coronary artery disease patients, the MMP-9 inhibitor curcumin was effective in preventing vascular diseases and suppressing MMP-9 expression [100]. An MMP-9-specific Mab, GS-5745, is undergoing a randomized phase 1 clinical trial for the treatment of ulcerative colitis after in vivo animal studies [101]. The MMP inhibitor COL-3 has entered in vivo phase 1 clinical trials, demonstrating tolerability at tested doses in patients with refractory metastatic cancer [102]. Additionally, the MMP-9-specific Mab andecaliximab has reached an in vivo randomized phase 2 clinical trial, where it has shown the regression of gastric cancer [98]. Andecaliximab (ADX) (CAS. 1518996-49-0) has been documented in detail regarding its in vivo clinical trial progression and characteristics. It demonstrated in vivo nanomolar potency (IC_50_) in phase 2 and 3 clinical trials concluded in 2019, although efficacy was not observed. It has been explored in combination chemotherapy regimens for gastrointestinal tract (GIT) inflammation, as well as gastric and gastroesophageal junction lesions. The non-selective MMP inhibitor, a low-molecular-weight hydroxamic acid (referred to as prinomastat, CAS. 192329-42-3), exhibited nanomolar drug potency (IC_50_) in vitro. During in vivo phase 3 clinical trials, concerns related to cytotoxicity emerged. The selective MMP-9 inhibitor (R)-ND-336 (CAS No. 2252493-33-5), which includes thiols and thiirane functionalities, has a Ki of 19 nM for MMP-9, a Ki of 127 nM for MMP-2, and a Ki of 119 nM for MMP-14. Being a covalent inhibitor, it selectively addresses diseases related to MMP-3, -9, and -24, and has been considered for combined use with gemcitabine. Among pan-MMP inhibitors, tetracycline (marketed as periostat, doxycycline hyclate) exhibits 2–5 µM potency (IC_50_). It is the only FDA-approved agent for periodontal inflammatory disease, and its effects have also been assessed under in vivo phase 4 clinical trials for multiple sclerosis. Another member of the tetracycline family with pan-MMP inhibitory activity, minocycline (CAS No. 10118-90-8), demonstrated 100–300 µM potency (IC_50_) in phase 4 clinical trials for in vivo stroke in cardiac atherosclerosis and is considered applicable to various diseases.

## 9. Conclusions

Earlier reviews primarily summarized the biological functions of MMPs and their inhibitors in inflammatory diseases and cancers [103,104,105,106,107,108,109]. Recently, there has been considerable interest in the molecular design and chemical synthesis of MMP inhibitors as well as the use of naturally occurring MMP inhibitors for biomedical applications. Currently, however, no effective MMP inhibitor has received FDA approval, as none have successfully completed the necessary clinical trials. Therefore, this review principally seeks to provide an overview of advanced discovery strategies for the design and development of future MMP inhibitors, emphasizing their therapeutic potential and subtype selectivity in cancer treatment. Future efforts will focus on low-molecular-weight compounds targeting MMP-9, which may protect against ischemic brain injury by inhibiting BBB disruption. Additionally, rather than broad-spectrum MMP inhibitors, the field is shifting toward the development of agents that precisely target specific MMP subtypes and engage pocket and exosite regions, rather than the active site itself. For MMP therapeutics undergoing clinical development, the current landscape for drug advancement remains unpromising. The present review highlights the currently characterized selective MMP inhibitors. Moreover, this review discusses the existing status and potential of MMP inhibitor development.

## Figures and Tables

**Figure 1 ijms-26-05553-f001:**
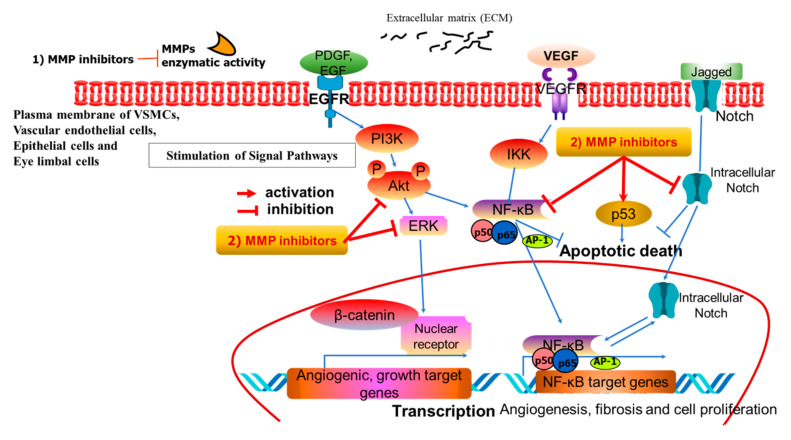
A representative pathway of MMP inhibitor-related intracellular signaling. MMP inhibitors exert their effects by targeting both kinase activity and MMP enzyme activity at the phosphorylation and enzyme levels, respectively. MMP inhibitors act in two different ways: (1) MMP enzymatic activity and (2) the stimulated signal pathways. MMP enzymes are associated with abnormal proliferation and remodeling in the present VSMCs, vascular endothelial cells, epithelial cells, and eye limbal cells.

**Figure 2 ijms-26-05553-f002:**
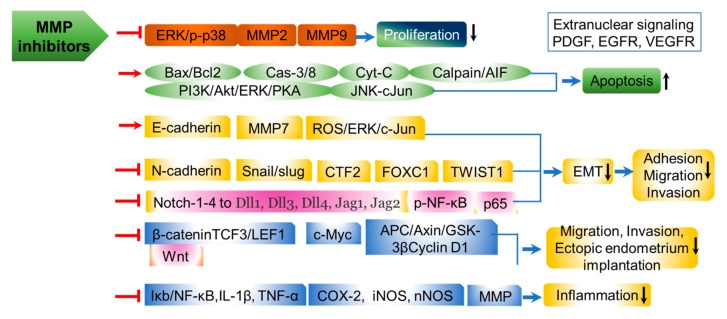
Schematic overview of the roles of MMPs and their inhibitors in regulating atherosclerotic development and progression, endometriotic development and progression in the uterine endometrium of women, fibrotic eye pterygium as a manifestation of fibrovascular fibrosis, and epithelial cell differentiation. MMP inhibitors, comprising TIMPs, natural products, synthetic compounds, semi-synthetic products, microRNAs, and monoclonal antibodies, affect various cellular processes related to vascular disease, endometriosis, pterygium fibrosis, and epithelial cell differentiation. Their functions include cerebrovascular protection, anti-angiogenesis activities, the regulation of fibrosis, and EMT. Specifically, efficacies cover vascular disease through mechanisms such as intimal thickening, the roles of endothelial and smooth muscle cells, orphan disease of fibrotic eye, pterygium mediated by limbic cells, TGF-β-induced endothelial-to-mesenchymal transition (EndMT), and the identification of biomarkers in epithelial cell differentiation. Additionally, they affect cancer behaviors such as initiation, cell cycle progression, proliferation, and metastasis, as well as fibrotic endometriosis involving β-catenin to 17β-estradiol (E2) and mucosal injury observed in laryngopharyngeal reflux.

## Data Availability

No new data were created or analyzed in this study.

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
