# Peer review of "Matrix Metalloproteinases and Their Inhibitors in the Pathogenesis of Epithelial Differentiation, Vascular Disease, Endometriosis, and Ocular Fibrotic Pterygium"

_ijms, 2025, doi:10.3390/ijms26125553_

Round 1

Reviewer 1 Report

Comments and Suggestions for Authors

The role of MMPs in pathophysiology is a broad field and the title also represents very broad set of topics that does not go well together. The authors are trying to review the role of MMPs in fibrosis in disease mainly. Perhaps the title should represent that and in the introduction this should be mentioned clearly. What is the common factor that led to select vascular, endometriosis and pterygium diseases as the target topic? 

While a lot of MMP substrates have been reported most of these substrates are only validated biochemically. The physiological relevance of these substrates has not been determined for many. Similarly, MMP inhibitors, as the reviewers have mentioned, are broad and not specific thus the use of these inhibitors can lead to undesired side effects.

Most of the studies mentioned here does not clarify if these studies are in vitro or in vivo studies. It would be great to have a mention of these. 

Comments on the Quality of English Language

Major - The document should be proof read a English should be fixed throughout the document.

Author Response

The role of MMPs in pathophysiology is a broad field and the title also represents very broad set of topics that does not go well together. The authors are trying to review the role of MMPs in fibrosis in disease mainly. Perhaps the title should represent that and in the introduction this should be mentioned clearly. What is the common factor that led to select vascular, endometriosis and pterygium diseases as the target topic?

   As suggested, the pathophysiological role of MMPs has been recognized for its broad involvement in cellular fibrosis remodeling. If remind, most studies of MMPs deal with cancer-associated invasiveness through the MMPs. However, the present manuscript focusses the fibrotic diseases, therefore, that seem not directly to be related together. That is the reason that the present title also represents broad topics due to their distinct properties. The present authors are highly interesting in the unrelated tissue remodeling of pathogenic epithelial differentiation, cardiovascular smooth muscle disease, endometriosis and eye pterygium. These representatively display the fibrotic tissue remodeling but not cancer-associated diseases, as we have reviewed the role of MMPs in fibrosis in disease mainly. We have revised the introduction section to justify the present title and incorporation of the different tissue remodeling. This makes it clear. The common factor MMP-based remodeling in vascular, endometriosis and pterygium diseases is timely the target topic.

While a lot of MMP substrates have been reported most of these substrates are only validated biochemically. The physiological relevance of these substrates has not been determined for many. Similarly, MMP inhibitors, as the reviewers have mentioned, are broad and not specific thus the use of these inhibitors can lead to undesired side effects.

As suggested, the issue of MMP substrates are still unsolved in the studies on pathophysiological MMPs. The reviewer raised the same issue occurred in the current drug development without therapeutic efficacy in vivo. Such in vivo and in vitro efficacies have been revised.

Most of the studies mentioned here does not clarify if these studies are in vitro or in vivo studies. It would be great to have a mention of these. 

This is the same question as the above. The reviewer raised the same issue occurred in the current drug development without therapeutic efficacy in vivo. Such in vivo and in vitro efficacies have been revised.

Reviewer 2 Report

Comments and Suggestions for Authors

Dear authors of Review 1
Matrix metalloproteinases and their inhibitors in the pathogen- 2
esis of epithelial differentiation, vascular disease, endometrio- 3
sis and eye fibrotic pterygium 4
Jun-Young Park 1,2#, Yeonwoo Choi 1#, Hee-Do Kim 1, Han-His Kuo 3, Yu-Chan Chang 3* and Cheorl-Ho Kim 1

I have tried my best to understand the writing, but the unaccepted quality of the English language is preventing me and taking a lot of my time to do quality review.

This is the first time in my long career that I cannot proceede with the review, due to insufficient quality of the English language.

I would gladly continue if someoneprofessional would take care of your paper to review and correct the English language.

I already started to correct it in order to read it again to be able to do scientific part of the review process, as I already notices in the Introduction some important missing elements regarding the involvement of MMP in signalling pathways.

Could you please hand over to profesional service to do, as per example I have provided and attached in this file?

For example, instead of the current, unacceptable abstract, it should be rewritten at least like this:

< !--StartFragment -->

Here is a revised, more concise, and scientifically precise version of your paragraph:

Matrix metalloproteinases (MMPs) play a crucial role in extracellular matrix (ECM) remodeling by degrading ECM components in a controlled endoproteolytic manner. In addition to ECM degradation, MMPs also cleave plasma membrane proteins involved in signaling pathways and disease progression. Structurally, MMPs require metal ions such as Zn²⁺ for catalytic activity. ECM degradation by MMPs facilitates tissue growth, formation, elongation, and adaptation to new environments under both physiological and pathological conditions. The regulation of these processes is tightly controlled by cellular MMP enzymes. Current knowledge on MMPs primarily focuses on their biological functions and implications in cancer; however, studies addressing their roles in vascular disease, endometriosis, ocular fibrosis, epithelial differentiation, and MMP inhibitors remain limited. This review highlights the involvement of MMPs in vascular disease and endometriosis, particularly in cases where endometrial tissue grows outside the uterus. Furthermore, we discuss their contributions to ocular fibrosis and epithelial differentiation, emphasizing their roles in disease progression. Given their therapeutic potential in human diseases, a deeper understanding of MMPs could lead to advancements in clinical applications and treatment strategies. This review also addresses the development and efficacy of MMP inhibitors as therapeutic agents.

< !--EndFragment -->

For the moment I will assign to this paper "a major revision"

Kind regards

Comments on the Quality of English Language

It is mandatory to correct style and grammar of the English language as to secure clarity, understanding and proper flow of the text, which are currently missing, thus preventing understanding and fair review process.

Author Response

Matrix metalloproteinases and their inhibitors in the pathogenesis of epithelial differentiation, vascular disease, endometriosis and eye fibrotic pterygium

Jun-Young Park 1,2#, Yeonwoo Choi 1#, Hee-Do Kim 1, Han-His Kuo 3, Yu-Chan Chang 3* and Cheorl-Ho Kim 1

I have tried my best to understand the writing, but the unaccepted quality of the English language is preventing me and taking a lot of my time to do quality review.This is the first time in my long career that I cannot proceede with the review, due to insufficient quality of the English language.I would gladly continue if someone professional would take care of your paper to review and correct the English language.I already started to correct it in order to read it again to be able to do scientific part of the review process, as I already notices in the Introduction some important missing elements regarding the involvement of MMP in signalling pathways.Could you please hand over to profesional service to do, as per example I have provided and attached in this file?

For example, instead of the current, unacceptable abstract, it should be rewritten at least like this:

Here is a revised, more concise, and scientifically precise version of your paragraph:

Matrix metalloproteinases (MMPs) play a crucial role in extracellular matrix (ECM) remodeling by degrading ECM components in a controlled endoproteolytic manner. In addition to ECM degradation, MMPs also cleave plasma membrane proteins involved in signaling pathways and disease progression. Structurally, MMPs require metal ions such as Zn²⁺ for catalytic activity. ECM degradation by MMPs facilitates tissue growth, formation, elongation, and adaptation to new environments under both physiological and pathological conditions. The regulation of these processes is tightly controlled by cellular MMP enzymes. Current knowledge on MMPs primarily focuses on their biological functions and implications in cancer; however, studies addressing their roles in vascular disease, endometriosis, ocular fibrosis, epithelial differentiation, and MMP inhibitors remain limited. This review highlights the involvement of MMPs in vascular disease and endometriosis, particularly in cases where endometrial tissue grows outside the uterus. Furthermore, we discuss their contributions to ocular fibrosis and epithelial differentiation, emphasizing their roles in disease progression. Given their therapeutic potential in human diseases, a deeper understanding of MMPs could lead to advancements in clinical applications and treatment strategies. This review also addresses the development and efficacy of MMP inhibitors as therapeutic agents.

As suggested, I would like to appreciate the reviewer-1 for His(orher) careful reading and revision of Abstract. I am very sorry for the poor English usage and abstract. We have fully revised the original manuscript by an University Englsh editing service (Harisco CO.). The certificate has been uploaded.

Too long manuscript and four different topics are potentially difficult to incorporate into one logic title. I am sorry for the present difficulty in reading and reviewing. Studies of MMPs are in general focused on cancer biology, metastasis and invasiveness, as well known. However, the present manuscriFpt deals with different and heterogeneous pathophysiological role of MMPs in fibrosis. Fibrotic diseases seem not directly to be related together but epithelial differentiation, cardiovascular smooth muscle disease, endometriosis and eye pterygium are highly associated with MMPs due to the fibrotic tissue remodeling.

Round 2

Reviewer 1 Report

Comments and Suggestions for Authors

the request to mention if the studies were in vivo or in vitro is not as the same as the previous comment. It is more informative to mention if the specific study is performed in vitro, in vivo or ex vivo on text.

Author Response

Reviewer-1’s 2nd comments

the request to mention if the studies were in vivo or in vitro is not as the same as the previous comment. It is more informative to mention if the specific study is performed in vitro, in vivo or ex vivo on text.

   As suggested, the in vivo and in vitro trials and efficacies have been revised in the 6.6 and 7 sections. Most MMP inhibitors are in vitro effective but not clinically. Therefore, Mabs are in vivo being developed. This is a current situation in development of such low molecular compounds. 6.6. MMP-targeting miRNA and monoclonal antibody (Mab)-based MMP inhibitors MicroRNAs (miRNAs) belong to the class of non-coding RNAs which are transcribed from genes but are not translated into proteins, functioning entirely at the RNA level. miRNAs are single-stranded RNAs, approximately 22 nucleotides in length. miRNAs target tumor suppressor genes or their promoters to regulate tumor progression. For instance, MiR-205, miR-214, and miR-203 inhibit MMP-9 expression, angiogenesis, and EMT. Conversely, miR-106a-5p promotes MMP-9 expression as well as the migration and invasion of tumor cells [94]. Long non-coding RNAs (lncRNAs), such as LncRNA UCA1, which are over 200 nucleotides in length, contribute to disease progression by upregulating MMP-9 expression [95]. In contrast, LncRNA LINC0031 suppresses MMP-9 expression via the P13K/Akt signaling pathway [96]. Furthermore, levels of circular RNAs (circRNAs) such as CircRNA DOCK1, produced through exon circularization, are elevated in tumor cells and facilitate the accumulation of MMP-9 in cancers [97].MMP-9 or MMP-14-specific Mabs have been evaluated in cancers such as breast tumors, gastric and gastroesophageal junction adenocarcinoma, and in autoimmune disease like ulcerative colitis [98]. Unlike chemical compounds, Mabs can inhibit MMP enzymes more specifically and generally exhibit fewer side effects and lower cytotoxicity than other MMP inhibitors. However, limitations of Mabs include poor penetration into tumor cells and high production costs. While no selective ADAM9, ADAM10, or ADAM17 inhibitors have been developed to date, ADAM17-specific Mabs, such as D1(A12), have been created to target the catalytic site in in vitro and in vivo studies [99]. The Mab D1(A12).45 D1(A12) blocks the interaction between integrin and the disintegrin region of the ADAM17 protein in in vitro studies. Moreover, the MED13622 Mab also selectively binds to ADAM17 protein, inhibiting the growth of migratory tumor cells and SMCs in preclinical studies. An MMP-9-specific monoclonal antibody inhibitor, andecaliximab, has shown efficacy in clinical trials. 

  1. The representative development status under clinical trials of human MMP therapeutics Currently, several MMP-9 inhibitors have advanced to phase-1 and phase-2 clinical trials but have not demonstrated significant efficacy. In a randomized clinical trial involving coronary artery disease patients, the MMP-9 inhibitor curcumin was effective in preventing vascular diseases and suppressing MMP-9 expression [100]. An MMP-9-specific Mab, GS-5745, is undergoing a randomized phase 1 clinical trial for the treatment of ulcerative colitis after in vivo animal studies [101]. The MMP inhibitor COL-3 has entered in vivo phase 1 clinical trials, demonstrating tolerability at tested doses in patients with refractory metastatic cancer [101]. Additionally, the MMP-9-specific Mab andecaliximab has reached an in vivo randomized phase 2 clinical trial, where it has shown regression of gastric cancer [102]. A Mab, andecaliximab(ADX) (CAS. 1518996-49-0), has been documented with details regarding its in vivo clinical trial progression and characteristics. It demonstrates in vivo nanomolar potency (IC50) in Phase 2 and 3 clinical trials concluded in 2019, although efficacy was not observed. It has been explored in combination chemotherapy regimens for gastrointestinal tract (GIT) inflammation, as well as gastric and gastroesophageal junction lesions. The non-selective MMP inhibitor, a low molecular weight hydroxamic acid (referred to as prinomastat, CAS. 192329-42-3), exhibited nanomolar drug potency (IC50) in vitro. During in vivo Phase 3 clinical trials, concerns related to cytotoxicity emerged . The selective MMP-9 inhibitor(R)-ND-336 (CAS No.2252493-33-5), which includes thiols and thiirane functionalities, has a Ki of 19 nM for MMP-9, a Ki of 127 nM for MMP-2, and a Ki of 119 nM for MMP-14. Being a covalent inhibitor, it selectively addresses diseases related to MMP-3, -9, and -24, and has been considered for combined use with gemcitabine. Among pan MMP inhibitors, tetracycline (marketed as periostat, doxycycline hyclate) exhibits 2-5 mM potency (IC50). It is the only FDA-approved agent for periodontal inflammatory disease, and its effects have also been assessed under in vivo Phase 4 clinical trials for multiple sclerosis. Another member of the tetracycline family with pan MMP inhibitory activity, minocycline (CAS No. 10118-90-8), demonstrated 100-300 mM potency (IC50) in Phase 4 clinical trials for in vivo stroke in cardiac atherosclerosis and is considered applicable to various diseases.

Reviewer 2 Report

Comments and Suggestions for Authors

Dear authors, thank you for ameding your manuscript to be easly readable and understandable.

I agree with the auhtor explanation about „However, the present manuscript deals with different and heterogeneous pathophysiological role of MMPs in fibrosis. Fibrotic diseases seem not directly to be related together but epithelial differentiation, cardiovascular smooth muscle disease, endometriosis and eye pterygium are highly associated with MMPs due to the fibrotic tissue remodelling.” In their response to reviewer.

Now as everything is clearer, I would like to point out that remodelling of ECM and signalling within the vascular smooth muscle cells along with their mobility and proliferation to achieve so is also supported by subset of matrix metalloproteinases ADAMs  that I have not found in your paper  mentioned and could be briefly added  before or after lines 142-143.

More about ADAM 12 can be found here DOI: https://doi.org/10.2174/092986711796504709

Involvement of the ADAM 12 in Thrombin-Induced Rat's VSMCs Proliferation

Secondly, consider to include  in Figure 1 a cross talk between major teo pathways. Several candidates shown to mediate this cross talk at this figure

Not the least but also cytosolic phospholipase 2 (cPLA2) is linked to VSMCs proliferation https://www.sciencedirect.com/science/article/abs/pii/S1065699509000134.

Kind regards

Author Response

Dear authors, thank you for amending your manuscript to be easly readable and understandable.

I agree with the auhtor explanation about „However, the present manuscript deals with different and heterogeneous pathophysiological role of MMPs in fibrosis. Fibrotic diseases seem not directly to be related together but epithelial differentiation, cardiovascular smooth muscle disease, endometriosis and eye pterygium are highly associated with MMPs due to the fibrotic tissue remodelling.” In their response to reviewer.

Now as everything is clearer, I would like to point out that remodelling of ECM and signalling within the vascular smooth muscle cells along with their mobility and proliferation to achieve so is also supported by subset of matrix metalloproteinases ADAMs  that I have not found in your paper  mentioned and could be briefly added  before or after lines 142-143.

More about ADAM 12 can be found here DOI: https://doi.org/10.2174/092986711796504709

Involvement of the ADAM 12 in Thrombin-Induced Rat's VSMCs Proliferation

As suggested, I fully agree with the reviewer’s suggestion regarding the VSMC proliferation and consequent atherosclerosis. Therefore, I have added the following sentences”

Membrane-anchored ADAM proteases on the cell surface is involved in cell migration, inflammation, and proliferation through signaling activation. ADAM17, ADAM9, and ADAM17 interact with surrounded immune cells. For example, when ADAM17 is associated with the development of atherosclerosis, it promotes inflammatory responses and regulates cellular oxidative stress, consequently exacerbation of pathological changes [10]. However, the intracellular ADAM17 function and role in in vivo environment are not well explored in atherosclerosis and the atherosclerotic action of ADAM17 is not clear understood yet. Copper homeostasis has been suggested associated with the mechanism of atherosclerosis. For example, ADAM1 upregulates atherosclerosis through cellular copper homeostasis [11]. Excessive free copper ions can cause oxidative stress, and damage cell membranes, thereby impairing the function of vascular endothelial cells, increasing the absorption of cholesterol in the arterial wall and the release of inflammatory factors, thereby accelerating the development of atherosclerosis [11]. ADAM12 also involves in thrombin-exerted proliferation of VSMCs [13]

10] Tang J, Frey JM, Wilson CL, et al. Neutrophil and Macrophage Cell Surface Colony-Stimulating Factor 1 Shed by ADAM17 Drives Mouse Macrophage Proliferation in Acute and Chronic Inflammation. Mol Cell Biol. Sep 1 2018;38(17).

11] Su TA, Shihadih DS, Cao W, et al. A Modular Ionophore Platform for Liver-Directed Copper Supplementation in Cells and Animals. J Am Chem Soc. 2018;140(42):13764-13774.

12] Smiljanic K, Dobutovic B, Obradovic M, Nikolic D, Marche P, Isenovic ER. Involvement of the ADAM 12 in thrombin-induced rat's VSMCs proliferation. Curr Med Chem. 2011;18(22):3382-6. doi: 10.2174/092986711796504709.

Secondly, consider to include  in Figure 1 a cross talk between major teo pathways. Several candidates shown to mediate this cross talk at this figure

As suggested, the figure has been updated to incorporate the suggested comments. Thanks a lot for careful reading.

Fig. 1. A representative pathway of MMP inhibitors-related intracellular signaling.   MMP inhibitors exert their effects by targeting both kinase activity and MMP enzyme activity at the phosphorylation and enzyme levels, respectively. MMP inhibitors act in two different ways of 1) the MMPs enzymatic activity and 2) the stimulated signal pathways. MMP enzymes are associated with the abonormal proliferation and remodeling in the present VSMCs, vascular endothelial cells, epithelial cells and eye limbal cells.

Not the least but also cytosolic phospholipase 2 (cPLA2) is linked to VSMCs proliferation https://www.sciencedirect.com/science/article/abs/pii/S1065699509000134.

As suggested, the issue of PLA2 is important to release the eicosanoic acid metabolites as the inflammatory mediators. VSMC proliferation and intimal thickening/rupture/abnormal migration is mediated, although this is out of the present review. In this review, I have not added the cPLA2 engagement. Thank you.
